# Changes in cortisol awakening responses (CAR) in menopausal women through short-term marine healing retreat program with specific factors affecting each CAR index

**Yesol Moon**[1], **Eunil Lee**[1,2], **Seoeun Lee**[1]*, **Gwang-Ic Son**[1], **Hangjin Byeon**[2], **Hyang-Ree Shin**[3], **Baatar Bolortsetseg**[4]

**1** Department of Biomedical Science, Graduate School & Department of Preventive Medicine, College of Medicine, Korea University, Seoul, Republic of Korea, **2** Department of Public Health, Graduate School, Korea University, Seoul, Republic of Korea, **3** Department of Nursing, Graduate School, Korea University, Seoul, Republic of Korea, **4** Biomedical Research Centre, Ghent University Global Campus, Incheon, Republic of Korea

* leese1105@korea.ac.kr

## Abstract

Recent studies have reported that the cortisol awakening response (CAR) is associated with various health risks. The different indices used to represent the CAR include the average cortisol levels in the morning immediately after waking (AVE); the total area under the curve of cortisol levels with respect to ground (AUCg); and the area under the curve of cortisol levels with respect to increase (AUCi). However, it is unclear which physiological phenomenon each index reflects. This study investigated the factors, such as stress, circadian rhythm, sleep, and obesity, affecting the CAR through a marine retreat-based healing program in which the anticipated stress of the participants could be controlled to some degree. Fifty-one menopausal women in their 50s and 60s were included, who performed beach yoga and Nordic walking for four days at an uncontaminated beach. The baseline CAR indices showed that the AVE and AUCg were significantly higher in the high sleep efficiency group than in the low sleep efficiency group. However, the AUCi decreased substantially with increasing age. The changes in the AVE, AUCg, and AUCi were calculated through the program, and it was found that the AVE and AUCg increased significantly more in the obese group than in the normal and overweight groups. The obese group also showed significantly decreased serum triglyceride and BDNF (brain-derived neurotrophic factor) levels compared to the low BMI group. Thus, it was confirmed that AVE and AUCg reflected physiological phenomena affected by factors such as sleep efficiency and obesity, whereas the AUCi was affected by factors such as age. In addition, the marine retreat program can improve the low levels of CAR associated with obesity and aging.

## Introduction

Cortisol Awakening Response (CAR), a sharp increase in cortisol levels that occurs immediately after awakening from sleep in the morning, is associated with various health risks [1–3].

**Data Availability Statement:** The data underlying the results presented in the study are available on Figshare (DOI: 10.6084/m9.figshare.21904017.v1).

**Funding:** This work was supported by the Wando Marine Healing Blue Zone Creation Project funded by the Presidential Committee for Balanced National Development in South Korea. The funders had no role in study design, data collection and analysis, decision to publish, or preparation of the manuscript.

**Competing interests:** The authors have declared that no competing interests exist.

And aging is associated with decrease of CAR because of hypoactivity of neuroendocrine expression of the hypothalamic-pituitary-adrenal axis (HPA-axis) [4]. CAR is blunted also in people with chronic stress or burnout [5], and a decline in CAR reflects poor physical and mental health [6].

Improvement of CAR by exercise has been reported, even in the elderly population [6, 7]. Older adults performing aerobic exercise showed increased CAR [7]. Being overweight or obese is closely related to low-level physical activities [8], and exercise alters the CAR in in individuals with obesity [9]. However, the associations between obesity and CAR are not reported consistently; a study with adult participants reported that individuals with obesity exhibited lower CAR levels [10], while another study reported an increase in CAR in men with abdominal obesity. As there are few reports on CAR in females with obesity [11], it is necessary to evaluate the association between obesity and CAR, especially in women.

Several factors influence the CAR, including age, exercise, sleep, and stress, and it is well-known that anxiety about the future and acute stress are particularly influential [12]. However, reports of increased or decreased CAR in stressful settings do exist [12, 13]. Various stress factors in our everyday lives affect sleep, and the CAR has also been shown to be associated with sleep [14]. Previous epidemiological and experimental studies have shown inconsistent results involving both an increase and decrease of CAR in people with low sleep quality [15]. The factors affecting the CAR are interrelated; therefore, comprehensively studying CAR changes while controlling for each factor is necessary.

A forest or marine retreat-based healing program can be carried out in an environment where future-related stress is reduced to some degree. Therefore, it is advantageous to use a retreat program to investigate associations between CAR and other factors after minimizing stress. Although the age-associated decline in CAR has been shown to improve through physical activity, few studies have examined the effect of exercise as part of a forest or marine retreat-based healing program on improvements in CAR [7]. This study aimed to investigate the association between CAR and other factors and determine how the association changed during the program.

The participants of most studies on CAR are those with specific occupations (doctors, nurses, military officers) or chronic stress [16]. Studies on athletes with competition stress or patients with psychiatric diseases are also reported [5]. Moreover, studies on healthy children or adolescents are also reported [17]. However, studies for menopausal women experiencing rapid physical and psychological changes are scarce. Therefore, this study aimed to identify the major factors affecting the CAR, excluding stress factors, in menopausal women. We evaluated the effects of various factors such as age, obesity, sleep, and circadian rhythm on the CAR in menopausal women who participated in a marine healing retreat program, including beach yoga and Nordic walking.

## Materials and methods

### 1. Recruitment of subjects

The participants were recruited in three batches of 20 from February to June 2021. A total of 60 female participants in their 50s and 60s who were non-smokers and had no difficulty of exercising were recruited. The participants had never undergone menopause-related surgery and were in spontaneous menopause. Additionally, the participants were not taking any medications known to affect HPA axis function, including opiates, steroids, or antipsychotics associated with cortisol secretion [18]. One participant dropped out on the first day of the study. Eight women were excluded from the final analysis based on the MINI Patient Health Survey scores that suspected psychological diseases [19], resulting in a total of 51 participants. The

study procedures were approved by the institutional review board (IRB) of Korea University (KUIRB-2021-0094-06). Detailed objectives and procedures of the study were informed, and written consent was obtained from the participants.

## 2. Study protocol for exercise and measurements

The program duration was five nights and six days at an uncontaminated beach on Shinji Island in Wando-Gun, Korea. The participants performed yoga and Nordic walking as instructed by experienced trainers for approximately 2 hours each on the program's second day. To determine changes in the CAR before and after the program in the study, the participants' saliva was collected to measure cortisol levels on the first and last day. In terms of circadian rhythms, stress, and exercise habits (which are known to affect the CAR), surveys and tests were conducted before and after the marine retreat-based healing program.

## 3. Cortisol measurement

Saliva was collected four times at 15, 30, and 45 min immediately after waking up in the morning [20]. For saliva collection, an investigator delivered tubes directly to the subjects and collected the tube. Salivette Cortisol (Sarstedt-Germany) was used to collect saliva [21]. For the pre-treatment of saliva, Celec's [22] procedures were followed. Salivary cortisol concentration was analyzed using the Cortisol ELISA Kit (IBL, USA) according to the kit protocol. The CAR, AVE, AUCg and AUCi indices were evaluated [16].

## 4. Melatonin measurement

To measure melatonin levels, saliva was collected 8 p.m. and 10 p.m. on the first and last day of the program [23], using Salivette (Sarstedt-Germany). For the pre-treatment of saliva, Römsing's [24] procedures were followed. Salivary melatonin concentration was analyzed using Melatonin ELISA (saliva) Kit (IBL, UAS) according to the kit protocol.

## 5. Circadian rhythm analysis

Throughout the program, the heart rate was automatically monitored every 5 min using a Mi Band 5 (Xiaomi, China) [25]. The measured heart rate data were analyzed using a cosine wave function, and the best-fitting model was determined using the least squares regression method [26]. Using MATLAB software (MathWorks, USA), we extracted three variables reflecting circadian rhythms: amplitude, mesor, and acrophase [25].

## 6. Sleep analysis

Nocturnal sleep was assessed using E4 wristbands (Empatica, U.S.) worn by the participants, which were research devices for real-time physiological monitoring [27]. Total sleep time and efficiency were evaluated by applying the Cole-Kripke algorithm after extracting the values of movements during sleep using an accelerometer sensor [28].

## 7. Heart Rate Variability (HRV) measurement

For HRV testing, electrocardiographic (ECG) measurements were performed twice daily, in the morning and evening, using a sensor for ECG measurements [29]. Before measurement, the participants relaxed for at least 15 min, and AD8232 (Analog Devices) sensor chips with a 200 Hz sampling frequency were attached to both wrists and the right ankle to obtain the ECG readings [30]. Based on the obtained signal, HRV analysis was performed using MATLAB software (MathWorks, USA).

## 8. Blood sampling and analysis

Triglyceride (TG) and Brain-derived neurotrophic factor (BDNF) were assessed by measuring serum TG or BDNF levels of blood samples. Blood samples were collected from each participant using the SST tube in the fasting state on the morning of the first and last days of the program. Immediately after blood collection, the tube was mixed up and down 6 times, and then left standing at room temperature for 30 minutes. The tube was centrifuged at 3000 rpm for 10 minutes. Blood samples were sent to the LabGenomics Laboratory (Korea), stored at -80 degrees until analysis. TG (mg/dL) concentrations was measured using Enzymatic assay (ADVIA® Chemistry XPT System; Siemens, Munich, Germany). BDNF (pg/ml) concentrations was measured using Total BDNF Quantikine ELISA Kit (R&D Systems, USA) and SpectraMax 190 Microplate Reader (Molecular Devices, USA).

## 9. Intensity of exercise

To measure individual levels of physical fitness and the intensity of the exercises performed, heart rate monitoring was performed using Polar OH1 (Polar Electro Oy, Finland) [31], a heart rate monitor worn by participants. Changes in energy consumption and exercise intensity were analyzed by applying the Karvonen formula [32] to the measured heart rate.

## 10. Questionnaire

To identify the general characteristics of the participants, their occupation, educational background, drinking habits, and medical history were investigated. Ordinary physical activity levels were investigated using the Korean version of the International Physical Activity Questionnaire (IPAQ) [33] To exclude people with psychiatric diseases from the study participants, the Korean version of the Mini International Neuropsychiatric Interview (MINI) Patient Health Survey was administered [19]. To explore changes in mood state, an evaluation using the Korean edition of the Profile of Mood States (K-POMS) was performed [34]. The Kupperman index was used to assess menopausal symptoms [35]. The Korean version of the Pittsburgh Sleep Quality Index (PSQI-K) was also used [36].

## 11. Statistical analysis

A normality test was conducted on the data used in the analysis. If the data were not normal, they were changed to a normal distribution for analysis by performing a log transformation. The ANOVA or t-test was performed to determine differences in CAR associated with the factors, including individual characteristics. A paired t-test was performed to compare the changes in CAR before and after the marine retreat-based healing program. Multivariate regression analysis was performed using stepwise selection to determine the main factors affecting CAR. The SAS software (version 9.4) was used for statistical analyses.

## Results

To determine any differrences in the CAR levels measured before the program according to the individual characteristics of the participants, differences in the CAR AVE, AUCg, and AUCi were analyzed before the program, depending on age, residential area, BMI, blood pressure, female hormone levels, and medical history. All the CAR indices showed a decreasing trend among the older age groups. The AUCi for the CAR indices was significantly higher in participants aged ≤55 years than in the 55 to 60 years group and >60 years group (p = 0.02). However, there was no significant difference in the CAR indices between groups divided

**Table 1. Differences in CAR based on individual characteristics before the marine healing program.**

| Variable | | n | AVE | p | AUCg | p | AUCi | p |
|---|---|---|---|---|---|---|---|---|
| **Age** ≤55 | | 14 | 3.42 ± 0.38 | 0.15 | 7.25 ± 0.38 | 0.19 | 631 ± 415 | 0.02* |
| 55~60 | | 21 | 3.22 ± 0.31 | | 7.06 ± 0.33 | | 375 ± 366 | |
| >60 | | 16 | 3.17 ± 0.42 | | 7.00 ± 0.44 | | 209 ± 424 | |
| **BMI** Normal & Overweight (<25) | | 27 | 3.34 ± 0.37 | 0.09 | 7.17 ± 0.38 | 0.11 | 384 ± 417 | 0.86 |
| Obese (≥25) | | 24 | 3.17 ± 0.36 | | 7.00 ± 0.38 | | 406 ± 442 | |
| **WHR** Normal (<0.85) | | 23 | 3.29 ± 0.41 | 0.66 | 7.11 ± 0.42 | 0.74 | 479 ± 456 | 0.44 |
| Obese (≥ 0.85) | | 28 | 3.24 ± 0.35 | | 7.07 ± 0.37 | | 322 ± 390 | |
| **Blood Pressure** Normal | | 30 | 3.24 ± 0.36 | 0.28 | 7.08 ± 0.38 | 0.24 | 359 ± 410 | 0.38 |
| Pre-hypertension | | 19 | 3.24 ± 0.40 | | 7.06 ± 0.41 | | 404 ± 458 | |
| Hypertension | | 2 | 3.68 ± 0 | | 7.55 ± 0.04 | | 792 ± 680 | |
| **AMH Level** Low (<0.01) | | 14 | 3.29 ± 0.35 | 0.68 | 7.12 ± 0.37 | 0.68 | 473 ± 482 | 0.26 |
| Intermediate (= 0.01) | | 29 | 3.22 ± 0.38 | | 7.05 ± 0.39 | | 309 ± 389 | |
| High (≥0.02) | | 8 | 3.35 ± 0.42 | | 7.18 ± 0.44 | | 553 ± 419 | |
| **Physical fitness** <76% | | 22 | 3.22 ± 0.39 | 0.52 | 7.05 ± 0.41 | 0.48 | 279 ± 386 | 0.10 |
| ≥76% | | 29 | 3.29 ± 0.37 | | 7.12 ± 0.37 | | 477 ± 438 | |
| **Hypertension History** | No | 43 | 3.26 ± 0.37 | 0.86 | 7.09 ± 0.38 | 0.87 | 376 ± 397 | 0.49 |
| | Yes | 8 | 3.28 ± 0.44 | | 7.11 ± 0.43 | | 489 ± 570 | |
| **Diabetes Mellitus History** | No | 44 | 3.26 ± 0.38 | 0.97 | 7.09 ± 0.39 | 0.99 | 397 ± 434 | 0.90 |
| | Yes | 7 | 3.27 ± 0.38 | | 7.09 ± 0.39 | | 377 ± 389 | |
| **Hyperlipidemia History** | No | 39 | 3.27 ± 0.38 | 0.61 | 7.11 ± 0.39 | 0.61 | 398 ± 415 | 0.91 |
| | Yes | 12 | 3.21 ± 0.38 | | 7.04 ± 0.38 | | 382 ± 473 | |
| **Metabolic Syndrome History** | No | 32 | 3.26 ± 0.38 | 0.98 | 7.09 ± 0.40 | 0.95 | 392 ± 416 | 0.97 |
| | Yes | 19 | 3.26 ± 0.38 | | 7.09 ± 0.38 | | 397 ± 449 | |

Each bar represents the mean ± SD. p-values were obtained by student t-test or ANOVA.

*p-value<0.05. BMI: body mass index; WHR: Waist to Hip Ratio; AMH: Anti-mullerian hormone.

according to the residential area, BMI, blood pressure, female hormone levels, and medical history, except for age (Table 1).

Each CAR index was analyzed by grouping sleep-associated variables, including melatonin, sleep efficiency, and circadian rhythm variables. And the circadian rhythm was measured by the amplitude and phase of the cosine curve of heart rate. And there was no significant difference in sleep efficiency by age group (S1 Table). Each CAR index showed variable results by sleep efficiency factor. The high sleep efficiency group showed higher AVE and AUCg among the CAR indices (p = 0.02, p = 0.01). AUCi did not reveal a significant difference between the high and low sleep efficiency groups (Table 2). No significant difference in CAR indices was observed between the groups divided based on other factors except the HRV stress markers. However, HRV was not significantly associated with the CAR indices in the multivariate analysis. The multivariate regression analysis showed that the significant factors before the program were sleep efficiency to affect both AVE and AUCg (p = 0.03, p = 0.04; S2 and S3 Tables), and age to affect AUCi (p = 0.06; S4 Table).

There were no significant factors associated with CAR indices observed after the program. The participants' age, BMI, blood pressure, female hormone levels, and medical history were analyzed to investigate differences in AVE, AUCg, and AUCi after the program (Table 3). The results showed that AUCi increased significantly more in the group with a normal waist-to-hip ratio (WHR) than in the group with a high WHR (p = 0.02). However, in multivariate analysis, no factors significantly affected AUCi (S5–S7 Tables). And no variables showed

**Table 2. Differences in CAR based on multiple variables including circadian rhythms, sleep, stress, and physical activities before the program.**

| Variable | | n | AVE | p | AUCg | p | AUCi | p |
|---|---|---|---|---|---|---|---|---|
| **Circadian Rhythm** | | | | | | | | |
| Amplitude | High | 15 | 3.38 ± 0.34 | 0.14 | 7.22 ± 0.35 | 0.13 | 426 ± 375 | 0.72 |
| | Low | 18 | 3.21 ± 0.38 | | 7.04 ± 0.39 | | 366 ± 557 | |
| Phase | High | 18 | 3.31 ± 0.40 | 0.45 | 7.15 ± 0.42 | 0.40 | 381 ± 456 | 0.88 |
| | Low | 15 | 3.23 ± 0.38 | | 7.06 ± 0.37 | | 408 ± 516 | |
| **Melatonin before sleep** | | | | | | | | |
| Melatonin AUCg | High | 27 | 3.23 ± 0.40 | 0.49 | 7.05 ± 0.41 | 0.39 | 397 ± 476 | 0.95 |
| | Low | 24 | 3.30 ± 0.35 | | 7.14 ± 0.36 | | 390 ± 366 | |
| Melatonin AUCi | High | 25 | 3.26 ± 0.38 | 0.97 | 7.09 ± 0.40 | 0.93 | 341 ± 425 | 0.39 |
| | Low | 26 | 3.26 ± 0.38 | | 7.10 ± 0.38 | | 447 ± 426 | |
| **PSQI** | | | | | | | | |
| Good sleepers | | 12 | 3.41 ± 0.36 | 0.20 | 7.25 ± 0.40 | 0.24 | 552 ± 482 | 0.26 |
| Bad sleepers | | 33 | 3.24 ± 0.40 | | 7.06 ± 0.40 | | 323 ± 406 | |
| Sleep disorder | | 6 | 3.09 ± 0.13 | | 6.96 ± 0.18 | | 491 ± 387 | |
| **Sleep** | | | | | | | | |
| Sleep Efficiency % | High | 25 | 3.39 ± 0.37 | 0.02* | 7.22 ± 0.38 | 0.01** | 420 ± 478 | 0.69 |
| | Low | 26 | 3.14 ± 0.34 | | 6.96 ± 0.36 | | 370 ± 388 | |
| Total sleep time | High | 22 | 3.31 ± 0.40 | 0.40 | 7.15 ± 0.40 | 0.36 | 505 ± 466 | 0.10 |
| | Low | 29 | 3.22 ± 0.37 | | 7.05 ± 0.38 | | 307 ± 375 | |
| **Profile of Mood States** | High | 24 | 3.28 ± 0.42 | 0.70 | 7.12 ± 0.43 | 0.65 | 378 ± 439 | 0.81 |
| | Low | 27 | 3.24 ± 0.34 | | 7.07 ± 0.35 | | 408 ± 420 | |
| **Kupperman index** | Mild | 13 | 3.22 ± 0.40 | 0.88 | 7.04 ± 0.43 | 0.88 | 425 ± 489 | 0.78 |
| | Moderate | 21 | 3.28 ± 0.36 | | 7.10 ± 0.37 | | 344 ± 403 | |
| | Severe | 17 | 3.27 ± 0.40 | | 7.11 ± 0.40 | | 434 ± 418 | |
| **Stress (Heart Rate Variability)** | | | | | | | | |
| SDNN | High | 22 | 3.35 ± 0.41 | 0.15 | 7.19 ± 0.42 | 0.13 | 516 ± 497 | 0.08 |
| | Low | 29 | 3.19 ± 0.34 | | 7.02 ± 0.35 | | 305 ± 347 | |
| RMSSD | High | 27 | 3.27 ± 0.39 | 0.91 | 7.09 ± 0.41 | 0.95 | 506 ± 438 | 0.04* |
| | Low | 24 | 3.25 ± 0.36 | | 7.09 ± 0.37 | | 262 ± 375 | |
| TP | High | 25 | 3.29 ± 0.40 | 0.60 | 7.12 ± 0.41 | 0.61 | 530 ± 445 | 0.02* |
| | Low | 26 | 3.23 ± 0.36 | | 7.06 ± 0.37 | | 258 ± 363 | |
| LF/HF ratio | High | 18 | 3.30 ± 0.35 | 0.16 | 7.15 ± 0.36 | 0.17 | 535 ± 433 | 0.17 |
| | Middle | 26 | 3.30 ± 0.38 | | 7.12 ± 0.39 | | 341 ± 416 | |
| | Low | 7 | 3.01 ± 0.38 | | 6.84 ± 0.41 | | 218 ± 375 | |
| **Physical activities** | | | | | | | | |
| Vigorous | High | 26 | 3.18 ± 0.39 | 0.11 | 7.01 ± 0.41 | 0.12 | 361 ± 433 | 0.58 |
| | Low | 24 | 3.35 ± 0.35 | | 7.18 ± 0.35 | | 429 ± 422 | |
| Moderate | High | 24 | 3.23 ± 0.39 | 0.60 | 7.05 ± 0.41 | 0.50 | 352 ± 411 | 0.51 |
| | Low | 26 | 3.29 ± 0.37 | | 7.13 ± 0.37 | | 432 ± 441 | |
| Walking | High | 26 | 3.20 ± 0.43 | 0.24 | 7.03 ± 0.44 | 0.27 | 494 ± 476 | 0.08 |
| | Low | 24 | 3.32 ± 0.31 | | 7.15 ± 0.32 | | 286 ± 338 | |
| Total IPAQ | High | 21 | 3.26 ± 0.43 | 1.00 | 7.08 ± 0.46 | 0.87 | 469 ± 506 | 0.29 |
| | Low | 29 | 3.26 ± 0.34 | | 7.10 ± .34 | | 340 ± 354 | |

Each bar represents the mean ± SD. p-values were obtained by student t-test or ANOVA.

*p-value<0.05

**p < 0.01. PSQI: Pittsburgh Sleep Quality Index(scale for assessing sleep quality); Kupperman index: woman menopausal evaluation index; SDNN: Standard Deviation of NN interval; RMSSD: Root Mean Square of the Successive Differences; TP: Total power; LF/HF Ratio: A ratio of Low Frequency to High Frequency; IPAQ: The International Physical Activity Questionnaires(A scale that evaluates the level of physical activity in daily life)

**Table 3. Differences in CAR based on individual characteristics after the marine healing program.**

| Variable | | n | AVE | p | AUCg | p | AUCi | p |
|---|---|---|---|---|---|---|---|---|
| **Age** ≤55 | | 14 | 30.8 ± 5.2 | 0.18 | 1504 ± 392 | 0.11 | 479 ± 430 | 0.87 |
| 55~60 | | 21 | 30.1 ± 9.2 | | 1376 ± 417 | | 459 ± 451 | |
| >60 | | 16 | 25.7 ± 8.9 | | 1183 ± 420 | | 403 ± 371 | |
| **BMI** Normal & Overweight (<25) | | 27 | 28.5 ± 7.9 | 0.79 | 1344 ± 427 | 0.99 | 494 ± 445 | 0.34 |
| Obese (≥25) | | 24 | 29.1 ± 9.2 | | 1343 ± 428 | | 382 ± 368 | |
| **WHR** Normal (<0.85) | | 23 | 30.4 ± 9.3 | 0.24 | 1450 ± 480 | 0.12 | 602 ± 398 | 0.02* |
| Obese (≥ 0.85) | | 28 | 27.5 ± 7.6 | | 1260 ± 359 | | 321 ± 387 | |
| **Blood Pressure** Normal | | 30 | 29.7 ± 9 | 0.39 | 1365 ± 424 | 0.70 | 425 ± 448 | 0.58 |
| Pre-hypertension | | 19 | 26.6 ± 7.6 | | 1288 ± 449 | | 444 ± 365 | |
| Hypertension | | 2 | 32.9 ± 1.7 | | 1519 ± 84 | | 743 ± 266 | |
| **AMH Level** Low (<0.01) | | 14 | 27.2 ± 8.5 | 0.72 | 1246 ± 406 | 0.60 | 421 ± 450 | 0.86 |
| Intermediate (= 0.01) | | 29 | 29.2 ± 9.3 | | 1385 ± 474 | | 473 ± 397 | |
| High (≥0.02) | | 8 | 29.8 ± 4.9 | | 1368 ± 230 | | 389 ± 446 | |
| **Physical fitness** <76% | | 22 | 27 ± 8.9 | 0.20 | 1231 ± 417 | 0.09 | 369 ± 405 | 0.25 |
| ≥76% | | 29 | 30.1 ± 7.9 | | 1432 ± 414 | | 505 ± 416 | |
| **Hypertension History** | No | 43 | 29.2 ± 8.6 | 0.34 | 1365 ± 436 | 0.38 | 448 ± 438 | 0.91 |
| | Yes | 8 | 25.9 ± 6.9 | | 1211 ± 328 | | 428 ± 221 | |
| **Diabetes Mellitus History** | No | 44 | 28.4 ± 8.8 | 0.52 | 1334 ± 446 | 0.70 | 460 ± 418 | 0.53 |
| | Yes | 7 | 30.7 ± 5.6 | | 1402 ± 263 | | 353 ± 396 | |
| **Hyperlipidemia History** | No | 39 | 28.6 ± 8.6 | 0.89 | 1339 ± 439 | 0.89 | 414 ± 431 | 0.32 |
| | Yes | 12 | 29.1 ± 8.1 | | 1360 ± 381 | | 556 ± 334 | |
| **Metabolic Syndrome History** | No | 32 | 28.7 ± 9 | 0.94 | 1346 ± 464 | 0.96 | 431 ± 441 | 0.75 |
| | Yes | 19 | 28.9 ± 7.5 | | 1340 ± 353 | | 470 ± 368 | |

Each bar represents the mean ± SD. p-values were obtained by student t-test or ANOVA.

*p-value<0.05.

BMI: body mass index; WHR: Waist to Hip Ratio; AMH: Anti-mullerian hormone.

significant differences in CAR indices among the groups based on circadian rhythm, sleep, or stress variables (Table 4).

Changes in CAR indices through the program were analyzed based on the individual characteristics of the participants (Table 5). Among the CAR indices, the AVE and AUCg significantly increased in the obese group based on BMI or body fat than in the normal or overweight groups (p = 0.005, p = 0.01, p = 0.001). Multivariate regression analysis also showed that BMI significantly affected the changes in AVE and AUCg during the program (p = 0.002, p = 0.004; S8 and S9 Tables).

Differences in CAR during the program were explored by grouping variables related to circadian rhythms, sleep, stress, and physical activities. Changes in the CAR indices were also significantly associated with sleep efficiency only in univariate analysis (Table 6). The results showed that AVE and AUCg significantly increased in the low sleep efficiency group, which had shown low CAR indices before the program (p = 0.03). There was no significant difference in the changes in the CAR indices depending on the other factors. However, multivariate regression analysis showed that BMI was the factor that significantly affected changes in AVE and AUCg during the program and not sleep efficiency (S8 and S9 Tables). No factor that significantly influenced AUCi was found (S10 Table). In addition, it was confirmed that there was no significant difference in exercise intensity according to the BMI group (S11 Table).

**Table 4. Differences in CAR after the program based on multiple variables including circadian rhythms, sleep, stress, and physical activities.**

| Variable | | n | AVE | p | AUCg | p | AUCi | p |
|---|---|---|---|---|---|---|---|---|
| **Circadian Rhythm** | | | | | | | | |
| Amplitude | High | 15 | 29 ± 8 | 0.51 | 1429 ± 478 | 0.28 | 525 ± 375 | 0.88 |
| | Low | 18 | 27 ± 9 | | 1254 ± 418 | | 505 ± 357 | |
| Phase | High | 18 | 29 ± 9 | 0.69 | 1314 ± 441 | 0.76 | 473 ± 329 | 0.48 |
| | Low | 15 | 27 ± 8 | | 1364 ± 473 | | 567 ± 402 | |
| **Melatonin before sleep** | | | | | | | | |
| Melatonin AUCg | High | 27 | 28 ± 10 | 0.43 | 1308 ± 489 | 0.52 | 375 ± 441 | 0.20 |
| | Low | 24 | 30 ± 7 | | 1386 ± 336 | | 527 ± 370 | |
| Melatonin AUCi | High | 25 | 28 ± 9 | 0.37 | 1310 ± 476 | 0.56 | 393 ± 419 | 0.36 |
| | Low | 26 | 30 ± 8 | | 1380 ± 364 | | 502 ± 406 | |
| **PSQI** | | | | | | | | |
| Good sleepers | | 12 | 32 ± 8 | 0.29 | 1455 ± 391 | 0.37 | 456 ± 355 | 0.68 |
| Bad sleepers | | 33 | 28 ± 9 | | 1337 ± 454 | | 467 ± 425 | |
| Sleep disorder | | 6 | 25 ± 6 | | 1157 ± 261 | | 305 ± 493 | |
| **Sleep** | | | | | | | | |
| Sleep Efficiency % | High | 25 | 30 ± 7 | 0.49 | 1365 ± 311 | 0.74 | 443 ± 427 | 0.98 |
| | Low | 26 | 28 ± 10 | | 1325 ± 505 | | 446 ± 408 | |
| Total sleep time | High | 22 | 27 ± 7 | 0.29 | 1299 ± 431 | 0.51 | 511 ± 413 | 0.32 |
| | Low | 29 | 30 ± 9 | | 1379 ± 422 | | 393 ± 412 | |
| **Profile of Mood States** | High | 24 | 29 ± 9 | 0.91 | 1323 ± 396 | 0.76 | 466 ± 406 | 0.75 |
| | Low | 27 | 29 ± 8 | | 1360 ± 450 | | 428 ± 424 | |
| **Kupperman index** | Mild | 13 | 27 ± 9 | 0.33 | 1236 ± 419 | 0.25 | 369 ± 367 | 0.71 |
| | Moderate | 21 | 31 ± 8 | | 1460 ± 454 | | 492 ± 485 | |
| | Severe | 17 | 28 ± 8 | | 1278 ± 369 | | 445 ± 356 | |
| **Stress (Heart Rate Variability)** | | | | | | | | |
| SDNN | High | 22 | 31 ± 9 | 0.18 | 1462 ± 481 | 0.09 | 551 ± 470 | 0.12 |
| | Low | 29 | 27 ± 8 | | 1258 ± 360 | | 368 ± 355 | |
| RMSSD | High | 27 | 30 ± 9 | 0.22 | 1425 ± 469 | 0.14 | 470 ± 445 | 0.64 |
| | Low | 24 | 27 ± 7 | | 1248 ± 348 | | 415 ± 379 | |
| TP | High | 25 | 29 ± 9 | 0.63 | 1392 ± 460 | 0.43 | 449 ± 463 | 0.94 |
| | Low | 26 | 28 ± 8 | | 1295 ± 386 | | 441 ± 365 | |
| LF/HF ratio | High | 18 | 29 ± 8 | 0.46 | 1325 ± 347 | 0.30 | 488 ± 432 | 0.65 |
| | Middle | 26 | 30 ± 8 | | 1421 ± 456 | | 453 ± 415 | |
| | Low | 7 | 25 ± 10 | | 1153 ± 465 | | 325 ± 389 | |
| **Physical activities** | | | | | | | | |
| Vigorous | High | 26 | 27 ± 8 | 0.24 | 1298 ± 456 | 0.42 | 452 ± 405 | 0.89 |
| | Low | 24 | 30 ± 8 | | 1397 ± 384 | | 436 ± 431 | |
| Moderate | High | 24 | 29 ± 8 | 0.66 | 1343 ± 374 | 0.99 | 433 ± 408 | 0.85 |
| | Low | 26 | 28 ± 9 | | 1344 ± 475 | | 457 ± 425 | |
| Walking | High | 26 | 28 ± 8 | 0.53 | 1326 ± 424 | 0.77 | 443 ± 474 | 0.98 |
| | Low | 24 | 29 ± 9 | | 1361 ± 431 | | 447 ± 350 | |
| Total IPAQ | High | 21 | 28 ± 9 | 0.60 | 1328 ± 484 | 0.83 | 376 ± 428 | 0.32 |
| | Low | 29 | 29 ± 8 | | 1355 ± 382 | | 495 ± 401 | |

Each bar represents the mean ± SD. p-values were obtained by student t-test or ANOVA. PSQI: Pittsburgh Sleep Quality Index(scale for assessing sleep quality); Kupperman index: woman menopausal evaluation index; SDNN: Standard Deviation of NN interval; RMSSD: Root Mean Square of the Successive Differences; TP: Total power; LF/HF Ratio: A ratio of Low Frequency to High Frequency; IPAQ: The International Physical Activity Questionnaires(A scale that evaluates the level of physical activity in daily life)

**Table 5. Differences in the changes of CAR through the program based on individual characteristics of the subjects.**

| Variable | | n | AVE | p | AUCg | p | AUCi | p |
|---|---|---|---|---|---|---|---|---|
| **Age** ≤55 | | 14 | 0.7 ± 9 | 0.17 | 47 ± 465 | 0.31 | -89 ± 396 | 0.32 |
| 55~60 | | 21 | 4.7 ± 8 | | 191 ± 394 | | 83 ± 522 | |
| >60 | | 16 | -0.2 ± 7 | | -13 ± 370 | | 181 ± 475 | |
| **BMI** Normal & Overweight (<25) | | 27 | -0.98 ± 8 | 0.005** | -47.11 ± 383 | 0.01* | 107 ± 472 | 0.55 |
| Obese (≥25) | | 24 | 5.61 ± 8 | | 249 ± 385 | | 24 ± 490 | |
| **Body Fat** Normal (<27%) | | 10 | -6 ± 7 | <0.001*** | -307 ± 384 | 0.001*** | 20 ± 500 | 0.69 |
| Overweight (27~33%) | | 10 | 1.5 ± 5 | | 86 ± 248 | | 185 ± 430 | |
| Obese (≥33) | | 31 | 5.1 ± 8 | | 221 ± 380 | | 47 ± 496 | |
| **WHR** Normal (<0.85) | | 23 | 3.3 ± 9 | 0.37 | 151 ± 441 | 0.31 | 168 ± 506 | 0.19 |
| Obese (≥ 0.85) | | 28 | 1.1 ± 8 | | 32 ± 379 | | -10 ± 448 | |
| **Blood Pressure** Normal | | 30 | 3 ± 9 | 0.26 | 126 ± 442 | 0.25 | 60 ± 513 | 0.91 |
| Pre-hypertension | | 19 | 1.5 ± 8 | | 73 ± 347 | | 98 ± 450 | |
| Hypertension | | 2 | -6.8 ± 1 | | -373 ± 13 | | -50 ± 333 | |
| **AMH Level** Low (<0.01) | | 14 | 0.5 ± 6 | 0.30 | -2.4 ± 304 | 0.24 | 10 ± 572 | 0.20 |
| Intermediate (= 0.01) | | 29 | 3.6 ± 9 | | 169 ± 410 | | 164 ± 430 | |
| High (≥0.02) | | 8 | -0.8 ± 10 | | -63 ± 516 | | -164 ± 433 | |
| **Physical fitness** <76% | | 22 | 0.9 ± 8 | 0.44 | 26 ± 392 | 0.38 | 80 ± 450 | 0.89 |
| ≥76% | | 29 | 2.9 ± 8 | | 131 ± 421 | | 62 ± 506 | |
| **Hypertension History** | No | 43 | 2.4 ± 9 | 0.45 | 102 ± 421 | 0.50 | 69 ± 487 | 0.97 |
| | Yes | 8 | -0.3 ± 6 | | -12 ± 329 | | 76 ± 460 | |
| **Diabetes Mellitus History** | No | 44 | 1.9 ± 8 | 0.78 | 79 ± 411 | 0.79 | 85 ± 487 | 0.58 |
| | Yes | 7 | 2.9 ± 9 | | 125 ± 420 | | -24 ± 446 | |
| **Hyperlipidemia History** | No | 39 | 1.3 ± 8 | 0.28 | 48 ± 411 | 0.23 | 11 ± 511 | 0.11 |
| | Yes | 12 | 4.4 ± 8 | | 218 ± 387 | | 272 ± 269 | |
| **Metabolic Syndrome History** | No | 32 | 2 ± 9 | 0.93 | 75 ± 427 | 0.81 | 34 ± 528 | 0.49 |
| | Yes | 19 | 2.2 ± 8 | | 105 ± 384 | | 132 ± 384 | |

Each bar represents the mean ± SD. p-values were obtained by student t-test or ANOVA.

*p < 0.05

**p < 0.01

***p < 0.001. BMI: body mass index; WHR: Waist to Hip Ratio; AMH: Anti-mullerian hormone.

## Discussion

The CAR is associated with various sociopsychological, physical, and mental health factors; therefore, studies on the CAR are being conducted in different fields [20]. However, there is controversy regarding the evaluation of CAR based on the average of 3 to 4 cortisol level measurements within an hour, immediately after awakening. Finally, a few indices were developed for CAR evaluation including AUCg, which represents total cortisol secretion; AUCi, which reflects dynamic changes of CAR; and AVE, which represents the average of cortisol measurements [37]. However, some studies have reported that each CAR index can be affected by various factors. In a study of patients with depression, a favorable prognosis for depression was anticipated through CAR AUCi; however, a relationship between depression and AUCg was not observed [38]. In addition, a study related to psychological stress and symptoms of depression reported blunting of AUCi [39]. A positive correlation between AUCg and physical activity has been previously reported [40]. This study showed that AVE and AUCg were affected by sleep and BMI, whereas AUCi showed a significant relationship with age. These results are

**Table 6. Differences in the changes of CAR based on multiple variables including circadian rhythms, sleep, stress, and physical activities.**

| Variable | | n | AVE | p | AUCg | p | AUCi | p |
|---|---|---|---|---|---|---|---|---|
| **Circadian Rhythm** | | | | | | | | |
| Amplitude | High | 15 | -0.3 ± 8 | 0.55 | -7.4 ± 392 | 0.67 | 98 ± 426 | 0.54 |
| | Low | 18 | 1.5 ± 9 | | 58 ± 451 | | 203 ± 516 | |
| Phase | High | 18 | -0.7 ± 9 | 0.29 | -68 ± 462 | 0.15 | 92 ± 446 | 0.41 |
| | Low | 15 | 2.6 ± 7 | | 151 ± 333 | | 234 ± 507 | |
| **Melatonin before sleep** | | | | | | | | |
| Melatonin AUCg | High | 27 | 2.4 ± 8 | 0.78 | 102 ± 365 | 0.77 | 10 ± 475 | 0.36 |
| | Low | 24 | 1.7 ± 9 | | 68 ± 460 | | 137 ± 483 | |
| Melatonin AUCi | High | 25 | 0.8 ± 8 | 0.28 | 29 ± 387 | 0.32 | 44 ± 500 | 0.70 |
| | Low | 26 | 3.3 ± 9 | | 146 ± 428 | | 97 ± 463 | |
| **PSQI** | | | | | | | | |
| Good sleepers | | 12 | 1.0 ± 10 | 0.87 | 21 ± 539 | 0.84 | -108 ± 507 | 0.08 |
| Bad sleepers | | 33 | 2.2 ± 8 | | 107 ± 385 | | 179 ± 421 | |
| Sleep disorder | | 6 | 3.1 ± 6 | | 92 ± 294 | | -187 ± 596 | |
| **Sleep** | | | | | | | | |
| Sleep Efficiency % | High | 25 | -0.6 ± 8 | 0.03* | -49 ± 429 | 0.03* | 69 ± 532 | 0.99 |
| | Low | 26 | 4.5 ± 8 | | 205 ± 355 | | 71 ± 436 | |
| Total sleep time | High | 22 | -0.4 ± 8 | 0.09 | -35 ± 399 | 0.07 | 48 ± 360 | 0.78 |
| | Low | 29 | 3.8 ± 8 | | 177 ± 398 | | 86 ± 556 | |
| **Profile of Mood States** | High | 24 | 1.2 ± 8 | 0.51 | 41 ± 396 | 0.50 | 138 ± 471 | 0.38 |
| | Low | 27 | 2.8 ± 9 | | 122 ± 421 | | 14 ± 486 | |
| **Kupperman index** | Mild | 13 | -0.1 ± 8 | 0.44 | -19 ± 444 | 0.43 | -57 ± 460 | 0.49 |
| | Moderate | 21 | 3.7 ± 8 | | 168 ± 399 | | 148 ± 522 | |
| | Severe | 17 | 1.8 ± 8 | | 62 ± 391 | | 70 ± 435 | |
| **Stress (Heart Rate Variability)** | | | | | | | | |
| SDNN | High | 22 | 2.7 ± 9 | 0.68 | 107 ± 434 | 0.76 | 80 ± 543 | 0.91 |
| | Low | 29 | 1.6 ± 8 | | 71 ± 396 | | 63 ± 438 | |
| RMSSD | High | 27 | 3.6 ± 8 | 0.17 | 159 ± 423 | 0.19 | -4.5 ± 419 | 0.25 |
| | Low | 24 | 0.3 ± 8 | | 3.7 ± 383 | | 154 ± 535 | |
| TP | High | 25 | 2.2 ± 8 | 0.90 | 92 ± 420 | 0.92 | -48 ± 436 | 0.09 |
| | Low | 26 | 1.9 ± 8 | | 80 ± 405 | | 183 ± 498 | |
| LF/HF ratio | High | 18 | 0.3 ± 8 | 0.53 | -25 ± 405 | 0.35 | -47 ± 548 | 0.40 |
| | Middle | 26 | 2.9 ± 9 | | 153 ± 421 | | 158 ± 445 | |
| | Low | 7 | 3.6 ± 8 | | 142 ± 357 | | 69 ± 384 | |
| **Physical activities** | | | | | | | | |
| Vigorous | High | 26 | 2.5 ± 9 | 0.69 | 99 ± 451 | 0.81 | 86 ± 515 | 0.81 |
| | Low | 24 | 1.5 ± 8 | | 71 ± 363 | | 52 ± 443 | |
| Moderate | High | 24 | 2.2 ± 8 | 0.87 | 96 ± 421 | 0.87 | 75 ± 452 | 0.95 |
| | Low | 26 | 1.8 ± 8 | | 76 ± 404 | | 65 ± 511 | |
| Walking | High | 26 | 2.9 ± 8 | 0.45 | 116 ± 401 | 0.61 | -12 ± 524 | 0.22 |
| | Low | 24 | 1.1 ± 9 | | 55 ± 421 | | 155 ± 419 | |
| Total IPAQ | High | 21 | 1.6 ± 10 | 0.79 | 63 ± 477 | 0.75 | -54 ± 457 | 0.13 |
| | Low | 29 | 2.3 ± 7 | | 102 ± 361 | | 155 ± 481 | |

Each bar represents the mean ± SD. p-values were obtained by student t-test or ANOVA.

*$p < 0.05$. PSQI: Pittsburgh Sleep Quality Index(scale for assessing sleep quality); Kupperman index: woman menopausal evaluation index; SDNN: Standard Deviation of NN interval; RMSSD: Root Mean Square of the Successive Differences; TP: Total power; LF/HF Ratio: A ratio of Low Frequency to High Frequency; IPAQ: The International Physical Activity Questionnaires(A scale that evaluates the level of physical activity in daily life)

consistent with Anderson and Wideman's report [41], in which each index of CAR may reflect different physiological phenomena.

In this study, the variable affecting AUCi before the program turned out to be age, while sleep efficiency had no significant relationship with AUCi, but a significant connection with AVE and AUCg. In addition, regarding changes in the CAR indices during the marine retreat-based healing program, the AVE and AUCg increased significantly in the obese group, while the increase in AUCi was not significant. The older adults suffer from poor sleep efficiency [42]; however, there was no age difference between the participants in the high and low sleep efficiency groups (S1 Table), which indicates that CAR is independently affected by age and sleep efficiency. Furthermore, it was found that BMI was a significant factor affecting both the AVE and AUCg. It has been reported that both AVE and AUCg are affected by exercise [7]. However, AUCi can be affected by age. A univariate analysis of the data before the program showed a significant difference in AUCi between the high and low RMSSD (root-mean-square of successive difference) groups and between the high and low TP (total power) groups. RMSSD and TP are HRV indices that are related to stress. Similar to previous studies, this result supports the possibility that AUCi is affected by psychological stress and depression [39]. However, our results showed factors influencing different CAR indices, which provide necessary information to understand the physiological phenomena that each index of CAR reflects.

With increasing age, the shape of the CAR curve flattens, indicating a decrease in both AUCg and AUCi [4]. Our results showed a decreasing trend of AVE, AUCg and AUCi in older participants. This is consistent with previous studies reporting that, with increasing age, CAR decreases due to hypoactivity of the HPA axis [4, 7]. Our results also indicate that the decrease in CAR with increasing age can be improved through a marine retreat-based healing program, and no significant difference was found between the old and young age groups. However, in our study, only AUCi showed statistical significance with respect to age, whereas AUCg did not show significance. It has been reported that total cortisol secretion during the day increases with age [43], but the diurnal amplitude of cortisol decreases [4]. This means that the baseline cortisol level is relatively high in older adults. Therefore, the AUCi index appears to be a sensitive marker of the neurophysiological status of older adults.

Sleep efficiency was the main factor that showed differences in AVE and AUCg before the marine healing program. According to the results of this study, the AVE and AUCg were equally low in the group with low sleep efficiency before the program. Similar to our results, it was reported that in law enforcement officers with low sleep quality, the AVE and AUCg were low [16]. And insufficient sleep also reduces CAR with the manipulation of the sleep environment [44]. BMI was also the primary factor affecting changes in the AVE and AUCg. The changes in CAR were not significant in participants with normal BMI, whereas AVE and AUCg greatly increased in the obese group with a BMI of 25 or higher. Few studies have investigated changes in CAR during the marine retreat-based healing program for individuals with obesity. There was no significant change in the BMI in obese and non-obese participants throughout the program, and there were no significant differences in exercise intensity based on heart rate measurements during the program (S11 Table). However, the TG and BDNF levels significantly decreased in the obese group after the program (p<0.05; S12 Table). Aerobic exercises reduce the TGs in adults who are overweight or have obesity [45], and exercising and dietary control have been shown to reduce BDNF in women with obesity [46]. While physical activity promotes health benefits, strenuous or prolonged exercise increases the production of reactive oxygen species in the blood and skeletal muscles [47]. Previous studies have reported that the circulating BDNF can act as a protective agent against oxidative damage; as oxidative stress increases and the BDNF levels decrease [48, 49]. In addition, a study reporting a decrease

in serum BDNF after 3 months of long-term exercise suggests the possibility that BDNF is reduced by exercise-induced mechanical stress and consumed to promote repair of peripheral nerves or tissues [50]. Our study has the limitation of not directly investigating changes in oxidative stress. However, through a mechanism similar to previous studies, our results suggest that serum BDNF has the potential to be used for protection and damage repair against exercise-induced oxidative stress through short-term exercise interventions.

In addition, several studies have reported decreased BDNF in patients with chronic stress or insomnia [51, 52]. However, our study showed no significant changes in stress markers (Table 6) with increase of CAR and decrease of BDNF. These findings suggested that the effect of exercise, independent of the stress factor, had a physiological effect to BDNF levels or CAR curves in separate ways. And the BDNF assay method used in our study mainly reflects mature BDNF in serum based on the report about the performance of ELISA kit analysis which showed a slight reactivity to pro BDNF [53]. Therefore, our results suggested that the BDNF changes through exercise program showed the consumption of mature BDNF. Thus, the retreat program seems to be effective in promoting changes of the CAR indices associated with changes of BDNF in obese subjects.

Currently with an aging population across the globe, among various methods to improve physical and mental health in the elderly, a forest healing is being touted as a healing program in the nature out of the urbanized environment. Physical activities in the natural environment have more positive effects on mental health than indoor activities. Therefore, it has been reported that the quality of life index in menopausal women who participated in forest healing retreat programs dramatically increased [51]. In addition to forest retreat-based healing, health promotion employing the marine environment has recently been growing. In this regard, the effects of a marine environment on mitigating psychological issues, including women's mood, depression, tension, and fatigue, were verified [54]. However, few studies have investigated the effects of marine retreat-based healing combined with exercise on the health of older adults. In this study, we observed the increase in the CAR of older adults who underwent a marine retreat-based healing program. Therefore, further comprehensive studies are needed to evaluate the significant factors for each CAR index changes to investigate the effects of marine retreat-based healing programs on improving indicators of physical and mental health, including CAR, in older adults.

There have been few studies for the associations between CAR and menopausal women through the marine healing program, which is recently known to have stress relaxation and health promotion effects [55, 56]. In the menopausal women in our study, there was no significant difference in CAR based on AMH (anti-Müllerian hormone) levels (Tables 1, 3, and 5) and menopausal symptoms (Kupperman index) (Tables 2, 4, and 6). A previous study reported a higher CAR in menopausal women experiencing menopausal symptoms more frequently [57], whereas another study reported that the CAR was blunted when the degree of menopausal symptoms was severe [3]. However, a relationship between CAR and menopausal symptoms in menopausal women was not observed in this study. Therefore, further studies on CAR in menopausal women are needed.

There are certain limitations to our study. The duration of the marine healing program was for a short period of six days and five nights. Moreover, the sample size for data analysis was small, including only 51 participants. However, our study identified the significant factors that cause changes in the CAR indices in menopausal women during the short-term exercise retreat program, which could have been stress factors. Further studies with larger populations are needed to validate the factors such as sleep efficiency, obesity and aging affecting each CAR index.

## Supporting information

**S1 Table. Differences in sleep efficiency based on age group before the marine healing program.** Each bar represents the mean ± SD. p-values were obtained by ANOVA.
(DOCX)

**S2 Table. Factors affecting AUCg before the marine healing program through multivariate regression analysis.** R2 = 0.23 Adjusted R2 = 0.16 p = 0.019*. p-values were obtained by multivariate regression analysis. *p-value<0.05.
(DOCX)

**S3 Table. Factors affecting AUCi before the marine healing program through multivariate regression analysis.** R2 = 0.16 Adjusted R2 = 0.09 p = 0.06. p-values were obtained by multivariate regression analysis.
(DOCX)

**S4 Table. Factors affecting AVE after the marine healing program through multivariate regression analysis.** R2 = 0.07 Adjusted R2 = -0.01 p = 0.50. p-values were obtained by multivariate regression analysis.
(DOCX)

**S5 Table. Factors affecting AUCg after the marine healing program through multivariate regression analysis.** R2 = 0.10 Adjusted R2 = 0.02 p = 0.30. p-values were obtained by multivariate regression analysis.
(DOCX)

**S6 Table. Factors affecting AUCi after the marine healing program through multivariate regression analysis.** R2 = 0.03 Adjusted R2 = -0.05 p = 0.82. p-values were obtained by multivariate regression analysis.
(DOCX)

**S7 Table. Factors affecting changes in AVE through the marine healing program through multivariate regression analysis.** R2 = 0.22 Adjusted R2 = 0.16 p = 0.017*. p-values were obtained by multivariate regression analysis. *p-value<0.05;**p < 0.01.
(DOCX)

**S8 Table. Factors affecting changes in AUCg through the marine healing program through multivariate regression analysis.** R2 = 0.19 Adjusted R2 = 0.13 p = 0.028*. p-values were obtained by multivariate regression analysis. *p-value<0.05;**p < 0.01.
(DOCX)

**S9 Table. Factors affecting changes in AUCi through the marine healing program through multivariate regression analysis.** R2 = 0.14 Adjusted R2 = 0.08 p = 0.09. p-values were obtained by multivariate regression analysis.
(DOCX)

**S10 Table. Factors affecting AVE before the marine healing program through multivariate regression analysis.** R2 = 0.25 Adjusted R2 = 0.18 p = 0.012*. p-values were obtained by multivariate regression analysis. *p-value<0.05.
(DOCX)

**S11 Table. Differences in the exercise intensity depending on BMI groups in the marine healing program.** Each bar represents the mean ± SD. p-values were obtained by student t-test.
(DOCX)

**S12 Table. Differences in the changes of TG and BDNF depending on BMI groups through the marine healing program.** Each bar represents the mean ± SD. p-values were obtained by student t-test. *p-value<0.05;**p < 0.01. TG: triglyceride; BDNF: brain-derived neurotrophic factor.
(DOCX)

## Acknowledgments

We thank Dr. KyungJu Lee, Department of Women's Rehabilitation Medicine, National Rehabilitation Center, for helping us design the study and understand the characteristics of menopausal participants. And we would like to thank Dr. DaYeon Shin from the Department of Food and Nutrition, Inha University, who created an appropriate diet for the marine healing program. We are grateful to ChaeBin Lee, of the Korea Leaders Sports Association, who taught participants beach yoga. Lastly, we would like to thank KyungTae Kim, a Nordic walking expert from the Korea Nordic Walking Federation. We would like to thank Editage (www.editage.co.kr) for English language editing.

## Author Contributions

**Conceptualization:** Yesol Moon.

**Data curation:** Yesol Moon, Hangjin Byeon.

**Investigation:** Seoeun Lee, Gwang-Ic Son, Hangjin Byeon, Hyang-Ree Shin, Baatar Bolortsetseg.

**Methodology:** Hangjin Byeon, Baatar Bolortsetseg.

**Project administration:** Eunil Lee, Seoeun Lee, Gwang-Ic Son, Hyang-Ree Shin.

**Supervision:** Eunil Lee, Seoeun Lee.

**Validation:** Eunil Lee.

**Writing – original draft:** Yesol Moon.

**Writing – review & editing:** Yesol Moon, Eunil Lee, Seoeun Lee.

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
