## [Decision Letter · Decision Letter 0]

4 Dec 2022

PONE-D-22-22264Changes in cortisol awakening responses (CAR) in menopausal women through short-term marine healing retreat program with specific factors affecting each CAR parameterPLOS ONE

Dear Dr. Seoeun Lee,

Thank you for submitting your manuscript to PLOS ONE. After careful consideration, we feel that it has merit but does not fully meet PLOS ONE’s publication criteria as it currently stands. Therefore, we invite you to submit a revised version of the manuscript that addresses the points raised during the review process.

It is undoubtedly potentially a good work and one that addresses a multifactorial and therefore difficult topic. However, there are a number of considerations that need to be addressed in depth. Among these, in addition to what has been stated by the reviewers, I would like to highlight: (1) The wording as well as the level of English should be thoroughly revised (2) Clarifying the research objective (and hypothesis) in terms of the research question (3) The section on instruments, which should be briefly described and which involve their scores, should be deepened (4) The presentation of the results should be improved. (3) The presentation of results should be ordered, prioritised and logically summarised, as should the tables, which contain too much information and are therefore difficult to understand (and there are too many of them). (4) The discussion should be rephrased according to the objective and research question, as well as according to the description of the results.

We look forward to receiving your revised manuscript.

Kind regards,

Juan-Luis Castillo-Navarrete, Ph.D.

Academic Editor

PLOS ONE

https://journals.plos.org/plosone/s/fileid=ba62/PLOSOne_formatting_sample_title_authors_affiliations.pdf.

For additional information about PLOS ONE ethical requirements for human subjects research, please refer to http://journals.plos.org/plosone/s/submission-guidelines#loc-human-subjects-research3. We note that the grant information you provided in the ‘Funding Information’ and ‘Financial Disclosure’ sections do not match.

“This work was supported by the Wando Marine Healing Blue Zone Creation Project funded by the Presidential Committee for Balanced National Development in South Korea.

“This work was supported by the Wando Marine Healing Blue Zone Creation Project funded by the Presidential Committee for Balanced National Development in South Korea.

Reviewers' comments:

Reviewer's Responses to Questions

**Comments to the Author**

1. Is the manuscript technically sound, and do the data support the conclusions?

Reviewer #1: Partly

Reviewer #2: Yes

2. Has the statistical analysis been performed appropriately and rigorously? 

Reviewer #1: N/A

Reviewer #2: Yes

3. Have the authors made all data underlying the findings in their manuscript fully available?

Reviewer #1: Yes

Reviewer #2: Yes

4. Is the manuscript presented in an intelligible fashion and written in standard English?

Reviewer #1: No

Reviewer #2: Yes

5. Review Comments to the Author

Reviewer #1: With all due respect for the work of the authors, it is however my opinion that the MS, who deals with an important topic in the field of improving the quality of life, needs a profound reorganization. The introduction is unnecessarily verbose. It does not stimulate or justify interest in the topic right from the first sentences. The purpose of the research is not clear, nor the choice of the population is justified (the experimental group is not at all described in its clinical characteristics in the relative missing paragraph "study population", for example for the possible intake of drugs that could modify HPA axis activities; spontaneous/surgical menopause?)

The results paragraph should be rewritten by providing the reader with a clear and detailed explanation of what has resulted, and taking away the feeling of simply being sent back to reading what is reported in the table!

Discussion may be improved including possible mechanisms and including a comment/comparison with similar studies in the area.

Reviewer #2: Congratulations to the authors for make a detailed analysis of multiple variables that may affect changes in cortisol awakening responses (CAR) in menopausal women through short-term marine healing retreat program.

Statistical analysis is adequate for the study of the data available. It is necessary to review the paragraph that describes Table 3 in the results. The multivariate analysis of WHR on AUCi parameter is not found in supplementary table S9.

In the same paragraph, the description of table S2 that indicates the existence of significant differences between BMI values for the high, middle, and low AUCi groups is confusing, because the values 25±2.6; 24±2.5 and 25±2.7 for high, middle and low AUCi respectively, are very similar.

The authors present a comprehensive discussion of their results associated with the main findings of the study. Based on the changes in AVE and AUCg evidenced according to the BMI, it would be interesting for subsequent studies with larger populations to include the analysis of changes in oxidative stress markers.

6. PLOS authors have the option to publish the peer review history of their article (what does this mean?). If published, this will include your full peer review and any attached files.

Reviewer #1: No

Reviewer #2: No

---

## [Author Response · Author response to Decision Letter 0]

17 Jan 2023

Thank you for your revision request, and we have faithfully revised the manuscript and results. The answers to each request were uploaded as an "Response to reviewers" file. Once again, we would like to ask the editors and reviewers for a careful review.

---

## [Decision Letter · Decision Letter 1]

8 Feb 2023

PONE-D-22-22264R1Changes in cortisol awakening responses (CAR) in menopausal women through short-term marine healing retreat program with specific factors affecting each CAR parameterPLOS ONE

Dear Dr. Seoeun Lee,

Thank you for submitting your manuscript to PLOS ONE. After careful consideration, we feel that it has merit but does not fully meet PLOS ONE’s publication criteria as it currently stands. Therefore, we invite you to submit a revised version of the manuscript that addresses the points raised during the review process.

ACADEMIC EDITOR: Thanking you for the effort involved in revising your paper which deals with a very interesting subject. The new version of your paper has improved substantially, however there are certain aspects that need to be improved and/or clarified. In addition to what has been expressed by the reviewers, I would like to point out some key aspects.

In the paragraph starting on line 249, TG and BDNF are addressed, in this regard:

(1) No mention is made in the methodology about the measurement of TG and BDNF, even more so considering the difficulties involved in measuring the latter, therefore:

(1.1) How was peripheral BDNF measured? Serum? Plasma? Platelets?

(1.2) What were the conditions of peripheral blood sampling (a simple forced aspiration at the time of peripheral blood sampling is sufficient to degranulate platelets and thus increase "plasma" BDNF levels) (https://doi.org/10.1016/j.neurobiolaging.2004.03.002).

(1.3) Was anticoagulant used to receive the peripheral blood sample? Which one?

(1.4) How long were the collection tubes left for before centrifugation, how long were they centrifuged, at what speed?

(1.5) What technique was used to measure BDNF? ELISA? If so, does it adequately discriminate mature BDNF from proBDNF (http://dx.doi.org/10.1038/srep17989)?

(1.6) How were TGs determined? methodology? sampling conditions?

(2) It is hypothesised that the exercise programme would reduce BDNF levels, how is this explained? Even more so, when there is ample literature reporting increased BDNF levels as a result of physical exercise and cognitive training (https://doi.org/10.1038/s41380-019-0639-2 ; https://doi.org/10.3390/ijms22168814).

(3) In the supplementary tables the units of TG and BDNF are not indicated.

The above points are relevant to clarify, even more so when peripheral BDNF levels have been proposed as a way to assess the integrity of the HPA axis, given its inverse relationship with cortisol levels (http://dx.doi.org/10.1038/s41386-019-0391-y ; https://doi.org/10.3109/07853890.2015.1131327). Please ensure that your decision is justified on PLOS ONE’s publication criteria and not, for example, on novelty or perceived impact.

We look forward to receiving your revised manuscript.

Kind regards,

Juan-Luis Castillo-Navarrete, Ph.D.

Academic Editor

PLOS ONE

Reviewers' comments:

Reviewer's Responses to Questions

**Comments to the Author**

1. If the authors have adequately addressed your comments raised in a previous round of review and you feel that this manuscript is now acceptable for publication, you may indicate that here to bypass the “Comments to the Author” section, enter your conflict of interest statement in the “Confidential to Editor” section, and submit your "Accept" recommendation.

Reviewer #1: All comments have been addressed

Reviewer #2: All comments have been addressed

2. Is the manuscript technically sound, and do the data support the conclusions?

Reviewer #1: Partly

Reviewer #2: Yes

3. Has the statistical analysis been performed appropriately and rigorously? 

Reviewer #1: Yes

Reviewer #2: Yes

4. Have the authors made all data underlying the findings in their manuscript fully available?

Reviewer #1: Yes

Reviewer #2: Yes

5. Is the manuscript presented in an intelligible fashion and written in standard English?

Reviewer #1: Yes

Reviewer #2: Yes

6. Review Comments to the Author

Reviewer #1: I got to read the first version of the MS. I must admit that this second version of the manuscript has been greatly improved.

However, it still needs more care in the introduction/discussion/as well as in the abstract

There are many inaccuracies, with sentences left there in inappropriate places: results are still presented in the introduction or extensions of the research on absolutely different topics are hypothesized. I am referring to the possible involvement of the mechanisms underlying oxidative stress in the effects induced by marine living on CAR. Why not bring up the stimulating effect of the marine environment on the thyroid?

I want to clarify that, in my opinion, you have not studied three different types of CAR but rather have used different ways of mathematically expressing the physiological increase in cortisol, which is evident in the first hour after waking up. It is up to the authors to decide which method is best adapted to the proposed protocol and thus make it easier to read the work.

In addition, even before talking about further studies, the authors should still apply themselves to make this work intelligible and acceptable according to the standards set by PlosOne

Reviewer #2: I appreciate the effort in restructuring the manuscript to respond to the reviewers' comments. The structure of the new manuscript satisfies the requirements sent by the reviewers, however I suggest two minor revision:

- In p9 L181 include "normal" to compare with high:...more in the group with a "normal" waist-to-hip ratio (WHR) than...

- Considering the adjustments made in the manuscript , it seems appropriate to include in the results a brief description about the supplementary tables S10 y S11, in order to guide the reader in the subsequent discussion of the scopes of the study.

7. PLOS authors have the option to publish the peer review history of their article (what does this mean?). If published, this will include your full peer review and any attached files.

Reviewer #1: No

Reviewer #2: No

---

## [Author Response · Author response to Decision Letter 1]

24 Mar 2023

We really appreciate the comments of the editor and reviewers and found that they helped to improve our paper. We have revised the submitted revised manuscript again responded to the advices of the editor and the reviewers. Please review "Response to reviewers".

---

## [Editor Report · Decision Letter 2]

5 Apr 2023

Changes in cortisol awakening responses (CAR) in menopausal women through short-term marine healing retreat program with specific factors affecting each CAR index

PONE-D-22-22264R2

Dear Dr. Seoeun Lee,

We’re pleased to inform you that your manuscript has been judged scientifically suitable for publication and will be formally accepted for publication once it meets all outstanding technical requirements.

Kind regards,

Juan-Luis Castillo-Navarrete, Ph.D.

Academic Editor

PLOS ONE

Additional Editor Comments:

It is really gratifying to see how the writing has evolved and how it has improved substantially. Undoubtedly it is, in my opinion, a contingent issue, so I have no doubt that this paper will be a contribution to the scientific community.
---

## [Editor Report · Acceptance letter]

11 Apr 2023

PONE-D-22-22264R2 

Changes in cortisol awakening responses (CAR) in menopausal women through short-term marine healing retreat program with specific factors affecting each CAR index 

Dear Dr. Lee:

I'm pleased to inform you that your manuscript has been deemed suitable for publication in PLOS ONE. Congratulations! Your manuscript is now with our production department. 

Kind regards, 

on behalf of

Dr. Juan-Luis Castillo-Navarrete 

Academic Editor

PLOS ONE